# Kinetic Modeling of Hydrogen Production by Dehydrogenation of Polycyclic Naphthenes with Varying Degrees of Condensation

**DOI:** 10.3390/molecules27072236

**Published:** 2022-03-30

**Authors:** Alexander N. Kalenchuk, Leonid M. Kustov

**Affiliations:** 1Chemistry Department, Moscow State University, 1 Leninskie Gory, Bldg. 3, 119991 Moscow, Russia; akalenchuk@yandex.ru; 2N.D. Zelinsky Institute of Organic Chemistry RAS, 47 Leninsky Prosp., 119991 Moscow, Russia; 3Laboratory of Nanochemistry and Ecology, National University of Science and Technology MISiS, 4 Leninsky Prosp., 119049 Moscow, Russia

**Keywords:** hydrogen storage, polycyclic hydrocarbons, dehydrogenation, reaction order

## Abstract

The kinetics of reactions of dehydrogenation of polycyclic naphthenes (cyclohexane, decalin, bicyclohexyl, *ortho*-, *meta*-, and *para*-isomers of perhydroterphenyl) is modeled on the basis of a formal comparison of kinetic equations of the 1st and 2nd orders based on real experimental data. It is shown that the reaction of the 1st order is predominating in the series of cyclohexane–bicyclohexyl–perhydroparatherphenyl. For all other substrates, the probability of describing the reaction in accordance with the equation of the 2nd order increases markedly, and for trans-decalin it becomes the predominant form of describing the kinetics of the reaction.

## 1. Introduction

The modern transport systems are based on oil and gas fuel, the widespread use of which contributes significantly to air and environmental pollution. In this regard, many alternative solutions have been proposed recently to replace or modify conventional fuels aimed at reducing the environmental footprint. One of the most well-known concepts is the use of hydrogen gas as fuel and its combustion in an internal combustion engine or fuel cell. One of the reasons that hinder the practical implementation of this approach is the lack of modern systems capable of providing compact storage of hydrogen and its transportation, since hydrogen under normal conditions is a flammable gas. It is shown [1,2,3,4,5,6,7,8,9,10] that systems based on liquid organic hydrogen carriers (LOHC), in particular, polycyclic hydrocarbons (PH), are a safe and economical way of storing and transporting hydrogen. However, the efficiency of PH-based systems largely depends on the development of a highly active, selective, and stable catalyst. At the same time, analysis of the literature shows that the difficulty of achieving the equilibrium due to the reversible nature of the dehydrogenation reaction of polycyclic naphthenes causes great differences in the description of kinetic mechanisms, as well as the determination of the order of these reactions. Among the studied substrates, the kinetics of dehydrogenation is understood only for cyclohexane, in the description of which the 1st reaction order is established [11,12,13,14,15]. There is no reliable information in the literature for more complex naphthenic substrates. At the same time, understanding the kinetics of reactions helps to choose the right conditions for their implementation. This contributes to an increase in the efficiency of using LOHC for the purposes of storage and release of hydrogen. This work is devoted to kinetic modeling of reactions of hydrogen production by dehydrogenation of polycyclic naphthenes with different degrees of condensation (cyclohexane, bicyclohexyl, *cis*- and *trans*-isomers of decalin, *ortho*-, *meta*-, and *para*-isomers of perhydroterphenyl) at 300–340 °C and atmospheric pressure.

## 2. Results and Discussion

Analysis of the experimental data obtained showed that the products of complete hydrogenation of aromatic hydrocarbons used in this work (benzene, C_6_H_6_; naphthalene, C_10_H_8_; biphenyl, C_12_H_10_; *ortho*-, *meta*-, and *para*-isomers of terphenyl, C_18_H_14_) are the corresponding naphthenic substrates (cyclohexane, C_6_H_12_; decalin, C_10_H_18_; bicyclohexyl, C_12_H_22_; *ortho*-, *meta*-, and *para*-isomers of perhydroterphenyl, C_18_H_32_). At the same time, steric *cis*- and *trans*-isomers are formed for decalin and isomers of perhydroterphenyl during the formation of final polycyclic naphthenes, the quantitative ratio between the isomers is indicated in Table 1. It was also established that the rate of formation of *trans*-isomers exceeds the rate of formation of *cis*-isomers, as well as the preceding intermediate products of the reaction [8,9,10].

It is known that the catalytic reaction consists of a number of successive stages, such as adsorption, desorption, diffusion, dissociation, and the reaction itself. The reversible reactions of hydrogenation of polycyclic aromatic substrates with a different extent of condensation and the reactions of dehydrogenation of the corresponding polycyclic naphthenic molecules represent multi-stage processes with a general scheme: X-H_2n_ ↔ X + H_2n_. Since there are no objectively developed criteria for comparing the kinetics of such different substrates, the determination of the kinetic model of the reaction requires numerous experiments to establish the contribution of each of the components. At the same time, a formal comparison method is known, which is based on the selection of the kinetic equation that best corresponds to the experimental data obtained [16]. In this work, the values of the rate constant were determined by substituting experimental data of the substrate concentration dependence into the kinetic equations of the 1st and 2nd orders, most typical for the naphthene dehydrogenation reaction (Table 2).

Common features for the calculated values of the rate constants *k*(I) and *k*(II) were determined using the Pearson linear correlation coefficient (*R*_I_ and *R*_II_) [17]:*R*_X,Y_ = (*M*[*X,Y*] − *M*[*X*] × *M*[*Y*])/((√(*M*[*X*^2^] − (*M*[*X*])^2^) × (√*M*[*Y*^2^] − (*M*[*Y*])^2^))(1)
where *M* denotes the mathematical expectation.

Table 3 shows the parameters of cyclohexane dehydrogenation on a 3 Pt/C catalyst. The values of the correlation coefficients calculated on the basis of the data presented in Table 3 are equal to *R*_I_ = 0.9558 and *R*_II_ = 0.7717, respectively. The ratio between the values of *R*_I_/*R*_II_ = 1.24 indicates an obvious advantage in describing the kinetics of the cyclohexane dehydrogenation reaction using the 1st order equation, compared with the 2nd order equation, which is consistent with the literature data [6,7,8].

Figure 1 shows the temperature dependences of the dehydrogenation conversion of the investigated polycyclic naphthenic substrates on a 3 Pt/C catalyst. It is obvious that the presence of steric isomers with an increase in the condensation degree of the naphthenic substrates under study leads to an increase in the number of reaction routes compared with the hydrogenation process and, accordingly, to a difference in the characters of the temperature dependences of the dehydrogenation conversion. Thus, in the case of decalin, the conversion in the *cis*-isomer dehydrogenation in the temperature range 300–340 °C approaches the maximum values, while the highest conversion of *trans*-decalin in this range barely reaches 70%. At the same time, the calculation of the equilibrium constants of all elementary acts occurring during the dehydrogenation of decalin and perhydroterphenyl indicates that the *cis*-isomer has a tendency to transition to a more stable *trans*-isomer over the entire temperature range under study [18,19]. Taking into account the *cis-trans* transition during dehydrogenation of decalin, this leads to a decrease in the overall reaction rate and affects the completeness of hydrogen release. Because of the smaller differences in the structure of *cis*- and *trans*-conformations from linear-jointed molecules of perhydroterphenyl, dehydrogenation curves for their *ortho*-, *meta*-, and *para*-isomers in the investigated range of temperatures differ not so much [9,10].

Table 4 shows the values of the rate constants for the dehydrogenation of the studied polycyclic naphthenes calculated on the basis of equations given in Table 2. The values of the calculated correlation coefficients for each of the substrates are given in Table 5.

From the data given in Table 5, it can be seen that for all substrates, the values *R*_I_ and *R*_II_ calculated using the rate constants have signs of a strong correlation dependence: [*R*_XY_] > 0.7. At the same time, it is also clear that for all substrates studied in the work, the correlation coefficient R_I_, when describing the reaction kinetics using the 1st order equation, is higher than when describing the reaction kinetics using the 2nd order equation. This indicates the similarity of the mechanisms of dehydrogenation within the groups under consideration. In particular, for all the studied substrates, a predominantly terminal character of dehydrogenation was observed, when first of all the terminal cyclohexane cycles participated in the reaction and only then the cycles associated with them [4,5]. The exceptions are *trans*-decalin and, to a lesser extent, *cis*- and *trans*-isomers of perhydro-*orto*-therphenyl.

In the series of cyclohexane–bicyclohexyl–perhydro-*para*-therphenyl, there is an obvious predominance of the reaction of the 1st order. In the case of condensed decalin cycles, the presence of the *cis-trans* transition complicates the description of reactions: if the *cis*-isomer demonstrates features belonging to the group with the 1st reaction order, then for *trans*-decalin, the equation of the 2nd order becomes the predominant form of describing the reaction kinetics. This is probably due to the excessive accumulation of the *trans*-isomer from the *cis*-isomer during the dehydrogenation reaction [15].

For linearly connected molecules of perhydroterphenyl isomers, the effect of the *cis-trans* transition also occurs, but not as strong as for decalin. Thus, for *cis*-perhydro-*para*-terphenyl, there is an obvious predominance of the reaction of the 1st order. For *trans*-perhydro-*para*-therphenyl, the probability of the reaction scenario in accordance with the 1st and 2nd order equations is noticeably leveled, but unlike *trans*-decalin, some prevalence of the 1st order reaction remains. For *meta*- and *ortho*-isomers, this ratio is preserved, but the difference between the description of the reaction using the 1st and 2nd order equations is even more leveled, especially for perhydro-*ortho*-terphenyl, for which side condensation reactions into cyclic triphenylene derivatives were detected during dehydrogenation [20].

## 3. Materials and Methods

### 3.1. Methods of Conducting Catalytic Dehydrogenation Reactions

For the dehydrogenation reaction, substrates obtained by complete hydrogenation of the corresponding commercial aromatic hydrocarbons were used: benzene, 99.5% (Acros Organics, Geel, Belgium), biphenyl, 99% (Acros Organics), as well as a mixture of terphenyl isomers of the Santowax-R brand (11.03 wt. % *o*-C_18_H_14_, 59.22 wt. % *m*-C_18_H_14_ and 29.75 wt. % *p*-C_18_H_14_). Hydrogenation was carried out in a high-pressure autoclave PARR-5500 (Moline, IL, USA) with an internal volume of 600 mL at a temperature of 180 °C and a pressure of 70 atm. The reaction mass obtained after hydrogenation was carefully separated from the catalyst and analyzed. For dehydrogenation, substrates were used which had a selectivity for the main product of at least 99.5% and did not contain products of side reactions [9,10]. The completeness of the reaction was determined chromatographically.

The substrates completely saturated with hydrogen, i.e., the naphthenic compounds prepared by exhaustive hydrogenation of the aromatic substrates, were used for dehydrogenation, which was carried out in a flow-through catalytic reactor (LHSV = 1 h^−1^, *P* = 1 atm, *t* = 1 h). All communications of the setup were thermostated at temperatures of 90–120 °C. Saturated substrates in the liquid state were fed into the reactor using a high-pressure pump of the HPP 5001 type. At the exit of the reactor, hydrogen and reaction products were separated. Sampling for analysis was carried out every hour of the reaction.

A prepared sample of 3 wt. % Pt/C [14] was used as a catalyst for both hydrogenation and dehydrogenation reactions. Platinum was dispersed onto the surface of a carbon carrier of Sibunit (Omsk, Russia, bulk density = 0.62 g/cm^3^) by incipient wetness impregnation of the carrier with an aqueous solution of [H_2_PtCl_6_] (wt. % Pt = 36.3%) in accordance with the procedure [14]. The specific surface area of the catalyst was *S*_BET_ = 304 m^2^/g, the average particle size of Pt *d*(Pt) = 2–3 nm, the dispersion of Pt *D* = 49%, and the average pore size *R* = 4 nm. The catalyst was activated immediately before the reaction at a temperature of 320 °C in a hydrogen flow of 30 mL/min for 2 h.

### 3.2. Chromatographic Analysis

The products of hydrogenation and dehydrogenation reactions were analyzed using a CrystaLlux-4000M chromatograph with a flame-ionization detector using a ZB-5 (Phenomenex, Torrance, CA, USA) and TR-FFAP capillary columns (Thermo Scientific, Waltham, MA, USA). The analysis was performed in a programmable mode of 70–220 °C at a heating rate of 6 °C/min. For more detailed identification of semi-hydrogenated products and reaction by-products, separate liquid samples were analyzed using a FOCUS DSQ II chromate-mass-spectrometer (Thermo Fisher Scientific, Waltham, MA, USA) with a TR-5MS capillary column (Thermo, Waltham, MA, USA). The conversion in hydrogenation and dehydrogenation (*X*) was calculated by the formula: *X* = (c_0_ − c)/c_0_ × 100%, where c_0_ and c are the initial and final concentrations of the substrate.

The selectivity (*S*) was calculated by the formula: *S*(i) = ∑ c(i)/∑ c(k) × 100%, where ∑ c(i) and ∑ c(k) are the sums of concentrations of a group of products and all the products, respectively.

## 4. Conclusions

Thus, the modeling of the kinetics of dehydrogenation reactions of polycyclic naphthenes (cyclohexane, decalin, bicyclohexyl, *ortho*-, *meta*-, and *para*-isomers of perhydroterphenyl) demonstrates that their features are determined by the structure, configuration, and degree of condensation. The use of the modeling method based on a formal comparison of kinetic equations has shown that for all polycyclic naphthenic substrates studied, the equation of the 1st order best corresponds to the experimental data obtained. In the series of cyclohexane–bicyclohexyl–perhydro-*para*-terphenyl, the signs of the reaction of the 1st order are predominant. For all other substrates, the probability of a reaction scenario in accordance with the 1st and 2nd order equations is noticeably leveled. For *trans*-decalin, the predominant form of describing the reaction kinetics is the 2nd order equation, which is apparently due to the excessive accumulation of the *trans*-isomer via isomerization of the *cis*-isomer during the dehydrogenation reaction.

## Figures and Tables

**Figure 1 molecules-27-02236-f001:**
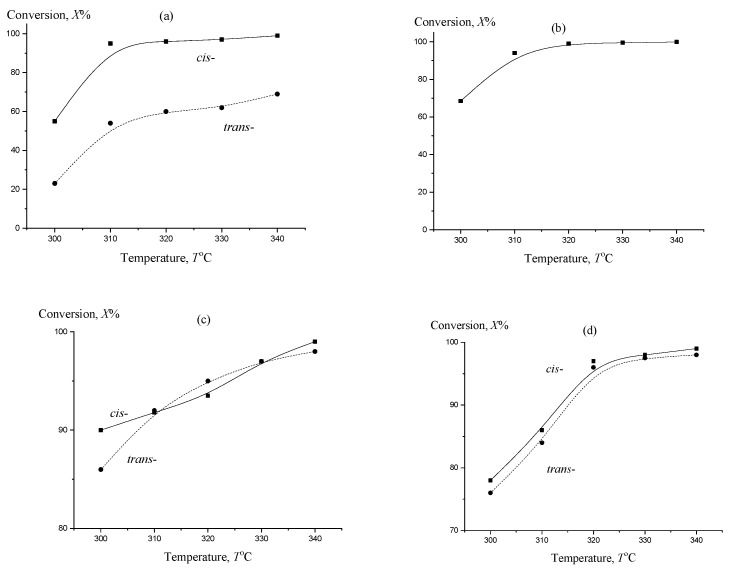
Temperature dependences of the dehydrogenation conversion of decalin: (**a**) bicyclohexyl; (**b**) perhydro-*para*-terphenyl; (**c**) perhydro-*meta*-terphenyl; (**d**) and perhydro-*orto*-terphenyl; (**e**) LHSV = 1 h^−1^, *P* = 1 atm, *t* = 1 h.

**Table 1 molecules-27-02236-t001:** Comparison of steric isomers of decalin and perhydroterphenyl (*C* is the content in the mixture).

Hydrocarbon	*Cis*-			*Trans*-
	*C*, %	*m. p.*, °C	*C*, %	*m. p.*, °C
Decalin	39	195	61	186
Perhydro-*o*-terphenyl	25	16–19	75	47
Perhydro-*m*-terphenyl	20	20–25	80	62
Perhydro-*p*-terphenyl	55	48	45	164

**Table 2 molecules-27-02236-t002:** Rate constant equations (*C*—concentrations of substrates; *C*_o—_initial; *C*—current).

Reaction Order	Formula for the Rate Constant
1	*k*_I_ = 1/t × ln(*C*_o_/*C*)
2	*k*_II_ = 1/t × (*C*_o_ − *C*)/(*C*_o_ × *C*)

**Table 3 molecules-27-02236-t003:** Parameters for the reaction of cyclohexane dehydrogenation (LHSV = 1 h^−1^, *P* = 1 atm).

	*T*, °C								
	250	270	280	290	300	310	320	330	340
*X*, %	29	40	55.8	62	81	82	97	98.4	99
*k*_I_, h^−1^	0.223	0.511	0.816	0.968	1.661	1.714	3.506	4.135	4.605
*k*_II_, h^−1^	0.025	0.007	0.013	0.016	0.043	0.046	0.323	0.615	0.990

**Table 4 molecules-27-02236-t004:** Parameters for dehydrogenation of naphthenic substrates (LHSV = 1 h^−1^, *P* = 1 atm).

Substrate		T, °C				
		300	310	320	330	340
Bicyclohexyl	*k*_I_, h^−1^	1.152	2.813	4.605	5.298	6.908
	*k*_II_, h^−1^	0.022	0.157	0.990	1.990	9.990
*cis*-Decalin	*k*_I_, h^−1^	0.800	3.000	3.219	3.507	4.605
	*k*_II_, h^−1^	0.012	0.190	0.240	0.323	0.990
*trans*-Decalin	*k*_I_, h^−1^	0.261	0.777	0.916	0.968	1.171
	*k*_II_, h^−1^	0.003	0.012	0.015	0.016	0.022
*cis*-Perhydro-*para*-terphenyl	*k*_I_, h^−1^	2.501	2.303	2.733	3.507	4.605
	*k*_II_, h^−1^	0.090	0.112	0.144	0.323	0.990
*trans**-*Perhydro*-para-*terphenyl	*k*_I_, h^−1^	1.966	2.526	3.000	3.507	3.912
	*k*_II_, h^−1^	0.061	0.115	0.190	0.323	0.490
*cis*-Perhydro-*meta*-terphenyl	*k*_I_, h^−1^	1.514	1.966	3.507	3.912	4.605
	*k*_II_, h^−1^	0.035	0.061	0.323	0.490	0.990
*trans*-Perhydro-*meta*-terphenyl	*k*_I_, h^−1^	1.427	1.833	3.219	3.689	4.200
	*k*_II_, h^−1^	0.032	0.053	0.240	0.390	0.657
*cis*-Perhydro-*ortho*-terphenyl	*k*_I_, h^−1^	1.897	1.833	1.772	1.743	1.715
	*k*_II_, h^−1^	0.057	0.053	0.049	0.047	0.046
*trans*-Perhydro-*ortho*-terphenyl	*k*_I_, h^−1^	1.599	1.619	1.660	1.687	1.714
	*k*_II_, h^−1^	0.040	0.041	0.043	0.044	0.046

**Table 5 molecules-27-02236-t005:** Empirical values of correlation analysis.

Substrate	Correlation Coefficients		
	*R* _I_	*R* _II_	*R*_I_/*R*_II_
Cyclohexane	0.956	0.772	1.24
*cis*-Decalin	0.924	0.881	1.05
*trans-*Decalin	0.929	0.963	0.97
Bicyclohexyl	0.992	0.822	1.21
*cis*-Perhydro-*para*-terphenyl	0.941	0.839	1.12
*trans*-Perhydro-*para*-terphenyl	0.979	0.975	1.02
*cis*-Perhydro-*meta*-terphenyl	0.979	0.948	1.03
*trans*-Perhydro-*meta*-terphenyl	0.978	0.968	1.01
*cis*-Perhydro-*ortho*-terphenyl	−0.981	−0.976	1.004
*trans*-Perhydro-*ortho*-terphenyl	0.996	0.996	1.00

## Data Availability

Not applicable.

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
