# Peer review of "Kinetic Modeling of Hydrogen Production by Dehydrogenation of Polycyclic Naphthenes with Varying Degrees of Condensation"

_molecules, 2022, doi:10.3390/molecules27072236_

Round 1

Reviewer 1 Report

No further revisions are required. 

Reviewer 2 Report

Dear Authors, Thank you for the opportunity to review your resubmission. After thoroughly reviewing the manuscript, the detailed revisions sufficiently addressed my previous critique. I have no further questions or concerns. I look forward to reading further work from your group.

Reviewer 3 Report

I agree with the publication in its current form.

This manuscript is a resubmission of an earlier submission. The following is a list of the peer review reports and author responses from that submission.

Round 1

Reviewer 1 Report

Summary- The kinetic modeling of the dehydrogenation reaction of various polycyclic naphthenes has been demonstrated. The predominance in first-order kinetics has been shown for cyclohexane, bicyclohexyl, and perhydroparatherphenyl. For ortho & meta perhydroterphenyl, the probability of second-order kinetics increases markedly. Whereas in the case of decalin, the second-order equation becomes predominant of the reaction kinetics, accumulation of the trans isomer being the reason for this observation. 

Edits-

Line 3: "varying degree of condensation" instead of varying the degree of condensation"

Line 57-58: citation should be added

Table 1 & 2- description of C, Co should be added

Line 69: "of" is misspelled

Suggestions-

1. The authors should mention the structures of substrates at least once in the manuscript.

2. The author should explain the contribution of this study to the development of LOHC based hydrogen production process with more clarity in the introduction of the manuscript (Line 31-34). 

Author Response

First, we are very grateful to the reviewer for the valuable comments and we revised the manuscript by taking into account all these comments and recommendation. Below we present the comment and the response:

Line 3: "varying degree of condensation" instead of varying the degree of condensation"

Response: The correction has been made.

Line 57-58: citation should be added

Response: We аdded citation [16]   

Table 1 & 2- description of C, Cshould be added

Response: In table 1, we added description of C. In table 2, we also added description of C, Co (initial and current concentrations) and corrected spelling of formulas 1 and 2

Line 69: "of" is misspelled

Response: the error has been corrected in the title of Table 1

Suggestions

  1. The authors should mention the structures of substrates at least once in the manuscript.

Response: In lines 47-50, the structural formulas are given after the names of the substrates

  1. The author should explain the contribution of this study to the development of LOHC based hydrogen production process with more clarity in the introduction of the manuscript (Line 31-34).

Response: After line 36, a sentence is added: “At the same time, understanding the kinetics of reactions helps to choose the right conditions for their implementation and, accordingly, increase the efficiency of using LOHC for the purposes of storage and release of hydrogen».

Best regards,

Authors Alexander N. Kalenchuk and Leonid M. Kustov

Reviewer 2 Report

This work details the dehydrogenation of polycyclic napthenes which after a quick search on Google Scholar, only a few other groups have worked on this interesting topic. The research methodology is appropriate, however, the manuscript doesn’t contain the bulk of the data/results used for preparing this paper. The substrates were first formed through hydrogenation, yet none of these experimental results are shown, or confirmation that pure substrates were produced. Please incorporate NMR and/or GC-MS verification of each substrates either in manuscript or Supplemental Information. There is also little detail on operation of the flow reactor such as flow rates, residence time, LHSV, etc.. Since the catalysts were made via incipient wetness instead of commercially purchased, there should be some detail on the characterization of the catalyst. While this manuscript wasn’t submitted to MDPI Catalysts, it would also be in the readers interest to see a general mechanism of the reactions. Overall I believe the work described in the manuscript is interesting, however, most of the work done isn’t in the paper.

Author Response

First, we are very grateful to the reviewer for the valuable comments and we revised the manuscript by taking into account all these comments and recommendation. Below we present the comment and the response:

Comment: The research methodology is appropriate, however, the manuscript doesn’t contain the bulk of the data/results used for preparing this paper.

Response: This work is based on the numerous experimental data obtained and published by the authors earlier. The major data on the preparation and characterization of the catalyst studied in this work are reported in our previous works. Part of the work is given in the list of references: [9], [10], [14], [15], [20]. The necessary data for calculations are presented in Figure 1.

Comment: The substrates were first formed through hydrogenation, yet none of these experimental results are shown, or confirmation that pure substrates were produced. Please incorporate NMR and/or GC-MS verification of each substrates either in manuscript or Supplemental Information.

Response: In this work, the products of hydrogenation and dehydrogenation reactions were analyzed using a CrystaLlux-4000M chromatograph with a flame-ionization detector using a ZB-5 (ZEBRON, USA) and TR-FFAP capillary columns (Thermo Scientific, USA). The analysis showed the absence of impurities of semi-hydrogenation products in the hydrogenated substrates. To further determine the purity of hydrogenated products, separate liquid samples were also analyzed using a FOCUS DSQ II chromato-mass-spectrometer (Thermo Fisher Scientific, USA) with a TR-5MS capillary column (Thermo, USA). For dehydrogenation, substrates were used without products of side reactions and with a selectivity for the main product of at least 99.5%. There was no need to use NMR and other methods to control the purity of the substrates. Hydrogenation of all the chosen aromatic substrates was also studied in our previous publications presenting the quantitative results of the hydrogenation reaction [9, 10, 14, 15, 20].

Comment: There is also little detail on operation of the flow reactor such as flow rates, residence time, LHSV, etc.

Response: Required data were added to section 3.1. Methods of conducting catalytic dehydrogenation reactions. Also, LHSV values are given in the tables.

Comment: Since the catalysts were made via incipient wetness instead of commercially purchased, there should be some detail on the characterization of the catalyst.

Response: The details of the characterization of the catalyst 3Pt/C have been published in the article [14] (Кalenchuk, А. N.; Bogdan, V. I.; Dunaev, S. F.; Кustov, L. М. Int. J. Hydrogen Energy 2018, 43, 6191.). There was no need to present the already published data in the manuscript or in the Supplementary Information.

Comment: While this manuscript wasn’t submitted to MDPI Catalysts, it would also be in the readers interest to see a general mechanism of the reactions.

Response: The general scheme of conjugated reactions of hydrogenation-dehydrogenation of substrates has been added to the text of the article.

Best regards,

authors Alexander N. Kalenchuk and Leonid M. Kustov

Reviewer 3 Report

In this manuscript, dehydrogenation of polycyclic naphthenes (cyclohexane, decalin, bicyclohexyl, ortho-, meta- and para-isomers of perhydroterphenyl) is modeled on the basis of a formal comparison of kinetic equations of the 1st and 2nd order based on real experimental data. Modeling of the kinetics of dehydrogenation reactions authors concluded that their features are determined by the structure, configuration and degree of condensation.

However, there are some issues that need to be resolved.

  1. On the page 2, the authors mentioned that: “It was also established that the rate of formation of trans-isomers exceeds the rate of formation of cis-isomers, as well as the preceeding intermediate products of the reaction”. Please add a reference
  2. At Figure 1 the reaction conditions must be mentioned.

Author Response

First, we are very grateful to the reviewer for the valuable comments and we revised the manuscript by taking into account all these comments and recommendation. Below we present the comment and the response:

However, there are some issues that need to be resolved.

  1. On the page 2, the authors mentioned that: “It was also established that the rate of formation of trans-isomers exceeds the rate of formation of cis-isomers, as well as the preceeding intermediate products of the reaction”. Please add a reference

Response: References [8-10] have been added

  1. At Figure 1 the reaction conditions must be mentioned.

Response: the reaction conditions have been added.

Best regards,

Authors Alexander N. Kalenchuk and Leonid M. Kustov